# PKM2 Regulates HSP90-Mediated Stability of the IGF-1R Precursor Protein and Promotes Cancer Cell Survival during Hypoxia

**DOI:** 10.3390/cancers13153850

**Published:** 2021-07-30

**Authors:** Han Koo, Sangwon Byun, Jieun Seo, Yuri Jung, Dong Chul Lee, Jung Hee Cho, Young Soo Park, Young Il Yeom, Kyung Chan Park

**Affiliations:** 1Personalized Genomic Medicine Research Center, Korea Research Institute of Bioscience and Biotechnology (KRIBB), Daejeon 34141, Korea; koohan@kribb.re.kr (H.K.); swbyun00@kribb.re.kr (S.B.); jieun527@kribb.re.kr (J.S.); jyuri3232@naver.com (Y.J.); dclee@kribb.re.kr (D.C.L.); cjh8308@gmail.com (J.H.C.); difco@korea.ac.kr (Y.S.P.); 2Department of Functional Genomics, University of Science and Technology, Daejeon 34113, Korea

**Keywords:** PKM2, IGF-1R, HSP90, hypoxia, survival

## Abstract

**Simple Summary:**

Generally, IGF-1R is overexpressed in most solid tumors, and its expression is significantly associated with poor prognosis in cancer patients. However, IGF-1R gene amplification events are extremely rare in tumors. It is, therefore, necessary to define the mechanism underlying IGR-1R overexpression to elucidate potential therapeutic targets. Our study, specifically, aimed to define the potential mechanisms associated with PKM2 function in regulating IGF-1R protein expression. PKM2 was found to be a non-metabolic protein that regulates HSP90 binding to and stabilizing the precursor IGF-1R protein, thereby promoting the basal level of mature IGF-1R protein. Consequently, *PKM2* knockdown inhibits the activation of AKT, a downstream effector of IGF-1R signaling, and increases apoptosis during hypoxia. Our findings reveal a novel mechanism for regulating IGF-1R protein expression, thus suggesting PKM2 as a potential therapeutic target in cancers associated with aberrant IGF signaling.

**Abstract:**

Insulin-like growth factor-1 receptor (IGF-1R), an important factor in promoting cancer cell growth and survival, is commonly upregulated in cancer cells. However, amplification of the *IGF1R* gene is extremely rare in tumors. Here, we have provided insights into the mechanisms underlying the regulation of IGF-1R protein expression. We found that PKM2 serves as a non-metabolic protein that binds to and increases IGF-1R protein expression by promoting the interaction between IGF-1R and heat-shock protein 90 (HSP90). PKM2 depletion decreases HSP90 binding to IGF-1R precursor, thereby reducing IGF-1R precursor stability and the basal level of mature IGF-1R. Consequently, *PKM2* knockdown inhibits the activation of AKT, the key downstream effector of IGF-1R signaling, and increases apoptotic cancer cell death during hypoxia. Notably, we clinically verified the PKM2-regulated expression of IGF-1R through immunohistochemical staining in a tissue microarray of 112 lung cancer patients, demonstrating a significant positive correlation (r = 0.5208, *p* < 0.0001) between PKM2 and IGF-1R expression. Together, the results of a previous report demonstrated that AKT mediates PKM2 phosphorylation at serine-202; these results suggest that IGF-1R signaling and PKM2 mutually regulate each other to facilitate cell growth and survival, particularly under hypoxic conditions, in solid tumors with dysregulated IGF-1R expression.

## 1. Introduction

Insulin-like growth factor (IGF) signaling is involved in the pathogenesis and progression of numerous malignancies. IGF-1 receptor (IGF-1R) activation results in the initiation of a signaling cascade involving kinases such as phosphatidylinositol 3-kinase (PI3K), protein kinase B (PKB/AKT), and mitogen-activated protein kinase (MAPK), which ultimately results in various cellular responses, including growth, transformation, development, and resistance to apoptosis [1,2,3]. In general, IGF-1R is overexpressed in most solid tumors, and its expression is significantly associated with poor prognosis in cancer patients [4,5,6]. The overexpression of IGF-1R has been implicated in cell proliferation, tumorigenesis, and apoptosis prevention induced by several agents [7]. Moreover, IGF 1/IGF 1R signaling protects cultured human cells from various injuries, including oxidative stress and hypoxia [8]. Therefore, targeting the IGF signaling system is considered a potentially promising strategy for cancer treatment.

Pyruvate kinase muscle type 2 (PKM2), an isoform of the pyruvate kinase enzyme that catalyzes the final and rate-limiting reaction in the glycolytic pathway, has been shown to promote cancer cell growth and survival [9,10]. PKM2 expression is highly upregulated in many cancers, including liver and lung cancers, and is associated with poor prognosis in cancer patients [11,12]. PKM2, the expression of which is regulated by microRNAs such as miR-338-3p and miR-199a, increases glycolysis and malignancy of hepatocellular carcinoma under hypoxic stress [13,14]. Meanwhile, there have been numerous reports on the non-enzymatic functions of the nuclear PKM2 dimer induced by the post-translational modifications of PKM2 protein [15]. Fibroblast growth factor receptor 1 induces the phosphorylation of tyrosine-105 in PKM2, resulting in a decrease in lactate production and promotion of tumor cell growth [16]. Stimulation of human glioblastoma cells by EGF causes phosphorylation of the serine (Ser)-37 residue in PKM2 and nuclear translocation of the PKM2 protein [17]. This, in turn, causes an increase in the expression of cyclin D1 and c-Myc, thereby promoting tumor cell growth and tumorigenesis. During hypoxia, the hypoxia-inducible factor 1 (HIF-1)-α transcriptionally upregulates PKM2, which binds to HIF-1-α and stimulates the transcription of glycolytic genes that promote glucose metabolism in cancer cells [18]. Previously, we reported that the phosphorylation of PKM2 at Ser-202 by AKT promotes the growth of cancers associated with aberrant IGF-1 signaling [19]. However, although most reports on the non-enzymatic functions of dimeric PKM2 have focused primarily on its nuclear function, a significant proportion of dimerized PKM2 remains cytosolic in cancer cells [20]. Considering that only one recent report has demonstrated the regulation of EGFR expression by cytosolic PKM2 regulating the HSP90–EGFR association [21], further investigations regarding the non-enzymatic oncogenic functions of cytoplasmic PKM2, are warranted. Low oxygen partial pressure or hypoxia occurs in most solid tumors due to rapid tumor growth and abnormal neovascularization [22]. Hypoxia exerts an intense selection pressure, inducing cell cycle arrest, as well as apoptotic and necrotic cell death in normal and neoplastic cells [23]. Conversely, it also promotes tumor propagation by aiding in the adaptation of cancer cells to nutritional deprivation or by aiding in their escape from the hostile environment through stimulation of the expression of glycolytic enzymes, angiogenic molecules, and survival factors, including growth factor receptors such as EGFR and IGF-1R [24,25]. Therefore, identifying factors and mechanisms responsible for controlling cancer cell survival under hypoxic conditions and inducing malignancy are vital to overcoming the current challenges facing cancer therapies.

In this study, we identified the association between PKM2 and the expression of IGF-1R protein in cancer cells, particularly under hypoxic conditions. Considering the induction of malignant phenotypes during hypoxic conditions, as well as the various oncogenic functions of PKM2 and IGF-1R signaling, it is highly valuable to define the oncogenic functions of IGF-1R-associated cytosolic PKM2 in the promotion of cancer cell survival during hypoxic conditions.

## 2. Materials and Methods

### 2.1. Reagents and Cells

The human Insulin-like Growth Factor-1 (Merck, Darmstadt, Germany, #I3769) was dissolved in PBS at concentration of 200 ng/mL. The HSP90 inhibitor NVP-AUY922 (Selleckchem, Houston, TX, USA, #S1069) and the proteasome inhibitors MG132 (Calbiochem, San Diego, CA, USA, #133407-82-6) and Bortezomib (Velcade, LC Laboratories, Woburn, MA, USA, #B-1408) were prepared in DMSO (Sigma-Aldrich, St. Louis, MO, USA, #D2650) at concentrations of 0.5 mM, 10 mM, and 10 mM, respectively. Cycloheximide (CHX, Sigma-Aldrich, 66-81-9) dissolved in DMSO was used at 50 ug/mL to block protein synthesis. Antibodies used in this study are listed in Appendix A.

Calu-3, A549, H23, H1299 (derived from human lung cancer), HeLa (derived from human cervix cancer), and HEK293T (derived from human embryonic kidney) cell lines were obtained from the American Type Culture Collection (ATCC, Manassas, VA, USA). Calu-3 and HeLa cell lines were cultured in Dulbecco’s modified Eagle’s medium (DMEM, Welgene, Gyeongsan, Republic of Korea, #LM001-05), while the A549 cell line was cultured in Roswell Park Memorial Institute 1640 medium (RPMI, Welgene, #LM011-01). All media were supplemented with 10% fetal bovine serum (GIBCO, Grand Island, NY, USA, #16000044) and 1% Antibiotics-Antimycotics (GIBCO, #15240-062). These cell lines were cultured every 2~3 days, at 80~90% confluence, to maintain continuous growth, and the cultured cells were incubated in 5% CO_2_ at 37 °C.

Hypoxic (1–3% oxygen) culture conditions were achieved with a hypoxia chamber (Sanyo, Osaka, Japan, MCO-5M incusafe) containing a gas mixture composed of 92–94% N_2_, 5% CO_2,_ and 1–3% O_2_.

### 2.2. cDNA and siRNA Transfection

pcDNA 3.1 (-) Myc-His-human PKM2, -mouse PKM2, pcDNA 3.1 Flag-human PKM2, -HSP90, pcDNA 3 HA-ubiquitin, and pBABE-bleo IGF-1R (Addgene, Watertown, MA, USA #11212) were used for overexpression experiments. The AccuTarget™ Negative Control siRNA (siCon, Bioneer, Daejeon, Korea), On-TARGET plus non-targeting pool siRNA, and custom-designed siPKM2 (Dharmacon, Lafayette, CO, USA, #D0018101050) were used in the gene-silencing study. The negative and non-targeting pool siRNAs had no significant homology to any known human gene sequences. For the transfection experiments, cDNA vectors and siRNAs were transfected using Lipofectamine^®^ 2000 reagent (Invtrogen, Carlsbad, CA, USA, #11668019) or Lipofectamine^®^ RNA-iMAX (Invitrogen, #13778075), respectively, according to the manufacturer’s instructions. Briefly, Lipofectamine^®^ reagents and cDNA or siRNA (20 nM) were mixed and incubated in Opti-MEM (GIBCO, #31-985-070) at room temperature for 15 min to form transfection complexes. Subsequently, the mixtures were added to each well, and then the cells were further incubated for 48h.

### 2.3. RT-PCR and Real-Time PCR

Total RNA was extracted using Tri reagent (Sigma-Aldrich, #T9424) according to the manufacturer’s instructions. A total of 5 µg of total RNA was used for cDNA synthesis; RT-PCR was conducted by VeritiPro™ Thermal Cycler (Applied Biosystems, Waltham, MA, USA, #A48141). As RT-PCR condition, Igf1r was amplificated by 25 PCR cycles at 60 °C of primer annealing temperature; Pkm2 and β-Actin were amplificated by 24 PCR cycles and 18 PCR cycles, respectively. Primer annealing temperature of those cDNAs was 58 °C in common. Quantitative PCR was carried out using CFX Connect Real-Time PCR Detection System (Bio-rad, Bio-Rad, Hercules, CA, USA, # 1855201). The reactions were conducted by triplicate in three independent experiments. The housekeeping gene β-Actin was used as an internal control to normalize target mRNA levels. Primers used in this study are listed in Appendix A.

### 2.4. Immunoprecipitation

For the immunoprecipitation (IP) study, HEK293T or calu-3 cells were transfected with siRNA targeting PKM2 and lysed in 500 μL RIPA lysis buffer. Following centrifugation for 15 min, the supernatants (~1–2 mg of total protein) were immunoprecipitated with 8 μL of IGF-1R antibody (Cell signaling, Danvers, MA, USA, #9750). This mixture was incubated overnight at 4 °C. Subsequently, 30 μL of protein agarose A/G beads (Santa Cruz, CA, USA, #sc-2003) were washed thrice using 500 μL RIPA lysis buffer. Following re-suspension in 100 μL of RIPA buffer, the beads were added to the mixtures and incubated again for 4 h at 4 °C. The formed immunocomplexes were washed thrice and disassembled by boiling with the SDS sample buffer. The samples were loaded in an SDS-polyacrylamide gel and immunoblotted using the appropriate antibodies. For the IP of Flag-tagged proteins, Flag-PKM2 or Flag-HSP90 were transfected in HEK293T, and the proteins were immunoprecipitated using anti-FLAG M2 affinity gel (Sigma, #A2220). The assay was carried out following the manufacturer’s protocol. All whole western blot figures can be found in the Appendix A.

### 2.5. Cell Viability

Cells were seeded at a density of 1 × 10^3^ cells/well in 96-well plates, and the siRNA for PKM2 was transiently transfected using Lipofectamine^®^ RNA-iMAX. Following incubation for 0, 1, 3, and 5 days, the cells were treated with 20 μL CellTiter-Blue^®^ (CTB, Promega, Madison, WI, USA, #G8080) reagent and incubated for 2 h at 37 °C in 5% CO_2_. Fluorescence was measured at an excitation wavelength of 530 nm and an emission wavelength of 590 nm. The number of cells exposed to hypoxic conditions was measured using an automated cell counter (TC-10, #1450009). The number of live cells was measured using the Trypan blue (GIBCO, #15250061) exclusion method. For live-cell count, the cells were initially seeded at a density of 1.3 × 10^5^ cells/well in 6-well plates and transfected with expression plasmids or siRNA for PKM2 or IGF-1R. Following incubation for 24 h, the cells were exposed to normoxic or hypoxic conditions. The cells were detached by trypsin treatment, centrifuged for 1 min, and re-suspended in 300 μL DMEM. Subsequently, 10 μL of the re-suspended cells were mixed with 10 μL of Trypan blue, and the mixtures were injected onto cell counting slides and counted using TC-10.

### 2.6. FACS Analysis

Calu-3 cells were transfected with siCon or siPKM2 for 24 h. Subsequently, they were cultured under normoxic or hypoxic (3% O_2_) conditions for 24 h. Following incubation, the cells were stained with FITC conjugated Annexin V and propidium iodide, according to the manufacturer’s instructions (BD Bioscience, Franklin Lakes, NJ, USA, #556547). Analysis of apoptotic cells was performed using the FACSVerse flow cytometer (BD Biosciences), and the data were processed using the BD FACSuite software (BD Biosciences).

### 2.7. Immunofluorescence

Calu-3 cells were seeded in an 8-well Nunc Lab-Tek Chamber Slide (Thermo Fisher Scientific, Waltham, MA, USA, #155411) and transfected with control siRNA or PKM2 siRNA. Following incubation for 48 h, the cells were fixed using 4% paraformaldehyde in PBS (pH 7.4), for 10 min, at room temperature. Subsequently, the cells were permeabilized by incubating in PBS containing 0.1% Triton X-100 (Biopure, Chungju, Republic of Korea, #9002-93-1) for 10 min. The samples were then blocked using 1% BSA in PBS-T for 30 min, and incubated with the appropriate antibodies overnight at 4 °C. On the following day, the samples were incubated with fluorescent conjugated secondary antibodies for 1 h at room temperature. Finally, the nuclei were counterstained with DAPI, and the samples were visualized using ZEISS LSM 900 with Airyscan 2 (Zeiss, Oberkochen, Germany).

### 2.8. Tissue Microarray and Immunohistochemistry

Tissue microarray (TMA) from 112 human lung adenocarcinoma patients was obtained from US Biomax, Inc (Derwood, MD, USA, HLugA120PG01). For immunohistochemical (IHC) analysis, we followed the manufacturer’s recommended protocol. Briefly, after deparaffinization and heat-induced epitope retrieval of the tissue sections, the slides were treated with 0.3% H_2_O_2_ in PBS for 15 min at room temperature to abolish endogenous peroxidase activity and incubated overnight at 4 °C with the primary antibodies. PKM2 and IGF-1R proteins were detected using the ABC avidin–biotin–peroxidase method (Vector Laboratories, Burlingame, CA, USA, #PK-4000). Nuclei were stained with hematoxylin, and samples were imaged with an ix71 (Olympus, Tokyo, Japan). IHC evaluation was performed by two scientists who provided a score for a tissue sample in which the intensity of the staining (negative staining = 0, very weak staining = 1, weak staining = 2, medium staining = 3, intense staining = 4, and very intense staining = 5) and the percentages of stained cells (0~100%) was multiplied. With this system, the maximum score was 500, and the minimum score was 0 (negative staining).

### 2.9. Statistical Analysis

GraphPad Prism 6 software was used for data analysis. The statistical significance was determined by one- or two-way ANOVA with Tukey post hoc tests for single or multiple comparisons, as appropriate. Wherever applicable, the assumptions of normality were verified using the Shapiro–Wilk test, Kolmogorov–Smirnov test, and D’agostino-Pearson omnibus test. *p*-values < 0.05 were considered statistically significant.

## 3. Results

### 3.1. Expression of PKM2 Is Associated with Cell Survival during Hypoxia

Prior to studying the mechanisms underlying the oncogenic functions of PKM2 during hypoxia, we compared the oncogenic activity of PKM2 in various cancer cells under normoxic and hypoxic conditions. Calu-3 cells expressing the siRNA of PKM2 were exposed to both normoxic and hypoxic conditions. Compared to the control, PKM2-downregulated cells showed a significant decrease in cell viability in both conditions. Interestingly, this reduction was more sensitive to the hypoxic condition (Figure 1A,C). Given the decrease in cell numbers over time, PKM2 depletion probably increased cell death rather than inhibiting cell growth during hypoxia. Similar effects on cell viability were observed during the downregulation of PKM2 in A549, H23, and H1299 cells (Figure 1B,C and Appendix A). To test if the reduction in cell viability caused by PKM2 deficiency correlated with apoptosis, we examined the effects of PKM2 knockdown on the activities of caspase 3/7 and the expression of apoptosis markers. Compared to the control exposed to hypoxia and the PKM2 knocked-down samples in normoxia, we found that the downregulation of PKM2 in Calu-3 cells significantly increased the activities of caspase 3/7 (Figure 1D) and the levels of apoptosis markers, cleaved caspase 9, and PARP (Figure 1E) during hypoxia. Similar effects on apoptosis were observed by the identification of cleaved PARP during the downregulation of PKM2 in A549 and H1299 cells (Appendix A). In addition, FACS analysis showed that PKM2 depletion in Calu-3 cells remarkably increased the number of apoptotic cells during hypoxia (Figure 1F). These results demonstrate that PKM2 is an important regulator of the hypoxic death of cancer cells.

### 3.2. PKM2 Depletion Attenuates the IGF-1R-AKT Signaling Pathway by Suppressing IGF-1R Expression

PKM2 expression is closely associated with hypoxia-induced apoptosis inhibition in various cancer cells (Figure 1). We had previously demonstrated that AKT physically interacts with and phosphorylates at Ser-202 residue of PKM2, promoting STAT5 activation and cancer cell growth [19]. These findings, together with the non-metabolic functions of PKM2, including the activation of HIF-1α activity via direct interaction and the phosphorylation of Histone H3 at Thr-11 [18,26], led to the estimation of the potential role of PKM2 in the regulation of AKT activation. We investigated the activation of AKT in PKM2-knockdown Calu-3 and A549 cells under hypoxic conditions. Interestingly, PKM2 depletion significantly reduced both levels of p-Thr308-AKT and p-Ser473-AKT (Figure 2A and Appendix A). Similar effects on AKT activation were observed during the downregulation of PKM2 in H1299 and H23 cells (Appendix A), suggesting the possible regulation of AKT signaling by PKM2. To verify the effect of PKM2 expression on AKT activation, we examined the role of PKM2 on the induction of the expression of IGF-1/AKT signaling-mediated anti-apoptotic genes (Bcl-2, Bcl2l1, and Mcl1) [27]. We confirmed that PKM2 depletion significantly reduced the induction of the downstream genes in the AKT signaling pathway during normoxia and hypoxia (Figure 2B and Appendix A).

Since we observed that the downregulation of PKM2 expression resulted in reduced AKT phosphorylation at both Thr-308 and Ser-473, we proposed the regulation of an AKT upstream factor by PKM2. To confirm this, we assessed whether the upstream factor IGF-1R was regulated by PKM2. Interestingly, we found that PKM2 depletion reduced IGF-1R protein level in various cancer cell lines, including A549, H1299, and H23 cells, without affecting the mRNA expression of IGF-1R (Figure 2C and Appendix A). A similar observation was made in the immunofluorescence study on the effect of PKM2 on IGF-1R expression (Figure 2D and Appendix A). IGF-1R silencing shows similar results compared to those obtained from PKM2 knockdown experiments with significantly reduced phosphorylation of AKT and cell viability, as well as markedly increased PARP cleavage during hypoxia (Appendix A). These results suggest that, during hypoxia, PKM2 deficiency led to reduced AKT activation and cell viability via the suppression of IGF-1R expression. Therefore, we further investigated whether restoration of PKM2 levels in PKM2-downregulated cells could restore IGF-1R expression and, subsequently, AKT phosphorylation. Results show that reconstitution of PKM2 resulted in an increase in IGF-1R and p-Ser473-AKT protein levels in PKM2-deficient Calu3, A549, and HeLa cells during both hypoxia and normoxia (Figure 2E, Appendix A), and that both the precursor and mature IGF-1R proteins gradually increased in a manner dependent on the amount of transfected PKM2 expression vector (Figure 2F and Appendix A). However, interestingly, PKM1 overexpression did not impact the expression of the precursor or mature IGF-1R protein (Appendix A). Furthermore, PKM2 mutants, low catalytic dimeric PKM2R399E, and catalytically dead PKM2K270M successfully restored IGF-1R levels in *PKM2* knockdown cells (Figure 2G and Appendix A), demonstrating that a non-metabolic function of PKM2 is involved in the regulation of IGF-1R expression. Together, these results suggest that PKM2 plays a crucial role in cancer cell growth and survival as a non-metabolic protein by promoting IGF-1R/AKT signaling.

### 3.3. PKM2 Physically Binds to and Promotes the Stability of IGF-1R Protein

Next, we sought to determine the mechanisms underlying the regulation of IGF-1R by PKM2. To investigate if the PKM2 regulation of IGF-1R expression is implicated at the transcriptional level, we measured the mRNA expression of IGF-1R and found that PKM2 knockdown does not affect the expression of IGF-1R mRNA during normoxia or hypoxia (Figure 2C,F, Appendix A). These results suggest that PKM2 regulates IGF-1R expression post-transcriptionally. Further, we observed that treatment with the proteasome inhibitors, MG132 and Velcade, restored the levels of IGF-1R protein and IGF-1-induced AKT phosphorylation in *PKM2* knockdown cells (Figure 3A and Appendix A). Moreover, when IGF-1R proteins were immuno-precipitated (IP) using an anti-IGF-1R antibody in cells exogenously expressing IGF-1R and HA-tagged ubiquitin (HA-Ub) (Figure 3B), or not (Figure 3C), the abundance of large molecule-sized (poly-ubiquitinated) IGF-1R proteins increased in the cells transfected with PKM2 siRNA compared to control cells. These results indicate that PKM2 deficiency leads to a significant increase in the ubiquitin/proteasome system (UPS)-dependent IGF-1R protein degradation.

Since it was identified that PKM2 regulates UPS-dependent degradation of IGF-1R protein and functions as a non-metabolic protein in the regulation of IGF-1R expression (Figure 2G), we next investigated if the PKM2 protein physically interacts with the IGF-1R protein. The PKM2 protein was Co-IP from the protein extract of HEK293T cells transduced with Flag-tagged PKM2 and IGF-1R genes and subjected to Western blot analysis using an anti-IGF-1R antibody. The result showed that the PKM2 protein binds to the IGF-1R proteins (Figure 3D and Appendix A). Besides, the reciprocal Co-IP experiment or endogenous IP using the anti-IGF-1R antibody revealed the presence of PKM2 in the IGF-1R precipitate (Figure 3E and Appendix A), confirming the physical interaction between the PKM2 protein and the IGF-1R proteins.

Interestingly, we found that the reconstitution of PKM2 proteins in the PKM2 knockdown cells successfully restored levels of both the precursor and mature IGF-1R proteins (Figure 2G). In addition, PKM2 binds not only to the mature form but to the precursor of IGF-1R proteins (Figure 3D). Together with the result of which PKM2 interaction to the mature IGF-1R markedly increases following the treatment of IGF-1, similarly to the interaction between PKM2 and EGFR, which requires EGFR activation (Figure 3D) [21], these results suggest the possibility that PKM2 could regulate the stability of both forms of the IGF-1R proteins. In particular, the result that PKM2 binds to the precursor IGF-1R suggests that PKM2 increases the basal level of the mature IGF-1R protein by regulating the IGF-1R precursor stability differently than how PKM2 regulates EGFR expression [21]. To determine which form of the IGF-1R proteins is regulated by PKM2, we measured protein stability using Cycloheximide in the PKM2 expression-regulated cells. Interestingly, the precursor form of IGF-1R protein was very fragile with a half-life of less than 30 min, in contrast to the mature form of which the level was sustained for 6 h after Cycloheximide treatment. In addition, ectopic expression of PKM2 significantly increased the half-life of the precursor IGF-1R (Figure 3F and Appendix A). These results suggest that basal expression of IGF-1R is regulated through PKM2-mediated stabilization of the precursor IGF-1R protein.

### 3.4. PKM2 Maintains IGF-1R Protein Stability by Mediating the Binding of IGF-1R to HSP90

HSP90, a molecular chaperone, reportedly regulates the protein folding and stability of various interacting factors, including certain oncogenic and tumor-suppressive factors [28]. HSP90 interacts with and promotes the expression of EGFR [21]. IGF-1R is also a target of HSP90. The blockade of HSP90 activity with HSP90 inhibitors leads to loss of IGF-1R protein stability [29]. Therefore, we assumed that HSP90 might also participate in the regulation of IGF-1R expression by PKM2. To investigate this hypothesis, we performed a Co-IP experiment to determine if IGF-1R and PKM2 bind to HSP90 in cancer cells. Interestingly, we found that PKM2, as well as IGF-1R, were identified in the sample precipitated with HSP90 (Figure 4A) and that the binding of IGF-1R to HSP90 was aborted by PKM2 depletion (Figure 4B,C and Appendix A). In particular, HSP90 binds more preferentially to the precursor IGF-1R protein, while PKM2 depletion markedly aborted this interaction (Figure 4C). These results suggest that PKM2 mediates the interaction between IGF-1R, particularly the precursor form, and HSP90 proteins.

To determine if PKM2 regulates the expression of IGF-1R protein mediated by HSP90, we assessed the stability of IGF-1R using Cycloheximide in PKM2 expression-regulated cells. The half-life of the precursor IGF-1R significantly increased following exogenous expression of HSP90 (Figure 4D and Appendix A); however, it significantly decreased (less than 30 min) following PKM2 depletion. Next, we investigated the effect of the HSP90 inhibitor, NVP-AUY922, on the expression of IGF-1R to identify the dependence on PKM2 for stabilizing IGFR protein by HSP90. Treatment with NVP-AUY922 reduced the increase in IGF-1R caused by overexpression of PKM2 in A549 cells (Figure 4E). Additionally, in Calu-3 cells, the inhibitor markedly reduced the basal level, as well as the PKM2 overexpression-induced level, of IGF-1R expression (Figure 4E). Similar effects were observed in H23 and H1299 cells (Appendix A). These results suggest that PKM2 mediates the stabilization of the IGF-1R precursor protein induced by HSP90. Consistent with these findings, the overexpression of PKM2 led to an increase in cell viability under hypoxic conditions. However, this effect was not observed in cells treated with NVP-AUY922 (Figure 4F and Appendix A). Hence, PKM2 contributes to maintaining IGF-1R expression and cancer cell viability by regulating the interaction between IGF-1R and HSP90.

### 3.5. PKM2 Expression Positively Correlated with IGF-1R Expression in Human Lung Adenocarcinoma Tissues

We next sought to clinically verify the regulatory effect of PKM2 on IGF-1R protein expression. To this end, we investigated the expression of PKM2 and IGF-1R with immunohistochemical staining of 112 human lung adenocarcinoma tissue microarrays (TMA) obtained from US Biomax, Inc. (Derwood, MD, USA) (Appendix A). The representative images depicting the patterns and strength of PKM2 and IGF-1R expression in lung cancer tissues are shown in Figure 5A. A significant correlation (R = 0.5208; *p* < 0.0001) was observed between the expression scores of PKM2 and IGF-1R in the lung cancer tissues (Figure 5B). Interestingly, the IGF-1R protein exhibited enhanced expression in cells preferentially expressing PKM2 (Figure 5C). Taken together, these data provide clinical evidence for a positive correlation between the expression of IGF-1R and PKM2, as well as the PKM2-mediated regulation of IGF-1R expression.

## 4. Discussion

IGF-1R signaling promotes malignant transformation and tumor progression in non-cancerous cells [30,31] and decreases tumor sensitivity to hypoxia, low pH, and low glucose environments [32,33,34]. AKT, the key factor in the IGF-1R signaling pathway, upon activation, promotes cell survival and tumorigenesis under hypoxic conditions [35,36]. Unlike other growth factor receptors, such as EGFR and HER-2, mutations activating the IGF-1R gene have rarely been reported in cancers. Additionally, gene amplification events are extremely rare in tumors [37]. However, IGF-1R is generally overexpressed in cancers and can be detected in most solid tumors [38]. During hypoxia, IGF-1R expression is increased, and the downstream effector AKT is activated, leading to hypoxic cancer cell growth and survival (Figure 2C). Despite extensive studies on IGF-1R signaling, the regulatory mechanisms of the IGF-1R protein are not well understood. Therefore, defining the mechanism by which IGF-1R expression is regulated in cancer cells will inform the design of cancer treatment aimed at targeting aberrant IGF signaling. We previously reported that AKT physically interacts with, and phosphorylates, PKM2 at the Ser-202 residue, thereby promoting IGF-1-stimulated cancer cell growth by activating the transcriptional activity of STAT5 [19]. Here, we have demonstrated the regulation of IGF-1R expression by PKM2, suggesting a positive feedback regulation of IGF-1R signaling by PKM2. Additionally, we have shown that PKM2 promotes cancer cell survival through the regulation of IGF-1R expression in hypoxic conditions. During hypoxia, the downregulation of PKM2 leads to the attenuation of IGF-1R signaling-mediated phosphorylation of AKT via suppression of IGF-1R protein expression and the promotion of hypoxia-induced apoptosis, suggesting that the regulation of IGF-1R signaling by PKM2 plays a crucial role in hypoxia-induced apoptosis. Epidemiological studies have shown that the IGF-1R signaling pathway disruption is strongly associated with the development of several common cancers, including prostate, breast, and colon cancer [39]. Thus, the IGF-1R-PKM2 signaling axis provides a potential therapeutic option, and its components serve as novel biomarkers in the treatment of several cancers, including lung cancer.

Although glycolysis is a low-efficiency process for ATP synthesis compared to OxPhos, cancer cells maintain a high level of glucose uptake and glycolytic activity, which is enhanced by various oncogenic events [40,41]. Glycolysis offers several advantages to cancer cells, including the ability to provide cells with substrates for the synthesis of nucleic acids and fatty acids required for increased proliferation. IGF-1R activation promotes mammary gland tumor development by increasing glycolytic activity and promoting biomass production [42]. Glucose metabolism in Trastuzumab-resistant breast cancer cells is upregulated by the stimulation of IGF-1R heterodimerization with ErbB receptors [43]. IGF-1R increases HIF-1-α expression by activating the AKT/mTOR pathway and subsequently increases the expression of glycolytic genes, known as HIF-1-α targets [44]. Many studies have reported that PKM2 also induces metabolic reprogramming by increasing cellular glycolysis. Although growth factor signaling-induced dimeric PKM2 has low pyruvate kinase activity and may reduce cellular glycolysis rates, PKM2 enhances glycolysis by increasing the expression of glycolytic genes via the transcriptional activation of HIF-1-α [18]. In addition to directly facilitating HIF-1-α functioning in the nucleus, our data suggest that PKM2 indirectly regulates metabolic reprogramming and survival by increasing IGF-1R protein stability and promoting IGF-1R signaling, indicating a positive feedback loop between IGF-1R signaling and PKM2 functioning. Moreover, the PKM2/IGF-1R signaling-mediated increase in glycolysis may contribute to the adaptive response of cancer cells to hypoxia.

Our study further revealed that PKM2 increases the cellular basal level of the mature IGF-1R protein by mediating the binding of HSP90, thereby increasing the stability of the precursor IGF-1R protein. Currently, HSP90 is considered a valuable target for cancer treatment. It has a variety of client proteins, including oncogenes such as EGFR and IGF-1R, and regulates their protein expression [21,29]. Treatment with HSP90 inhibitors significantly reduces client protein levels and inhibits embryonic stem cell proliferation and survival [45]. Treatment with anticancer drugs, such as Imatinib, increases HSP90 protein expression, thereby inducing resistance to the drug. However, HSP90 has been shown to be ubiquitously and highly expressed (approximately 1–2% of the total mammalian cellular proteins) in all types of healthy cells [28]. The efficacy of Hsp90 inhibitors in clinical studies is curtailed by their toxicity, which imposes an insufficient dose of the administered drug, leading to the lack of adequate inhibition of target proteins [46]. The other primary reason for the limited efficacy of Hsp90 inhibitors is the inevitable activation of HSF1, a transcription factor that causes heat-shock responses, such as induction of Hsp27, Hsp40, and Hsp70 to allow cells cytoprotective heat-shock responses [47]. These characteristics of HSP90 cause it to induce specific limited effects, thereby impeding the usage of HSP90 inhibitors [48], necessitating the use of other strategies such as blocking the interaction of HSP90 to its client proteins. PKM2 could provide the selectivity for the binding of HSP90 to its client proteins. HSP90 binds to mutant EGFR in a PKM2-dependent manner [21]. In our study, we found that PKM2 expression was necessary for the binding of HSP90 to IGF-1R, suggesting the target specificity imparted by PKM2 in regulating the expression of its oncogenic target proteins. Our results highlight the master role for PKM2 in the regulation of protein stability of the numerous oncogenic HSP90 client proteins and its potential use in the treatment of cancer by targeting HSP90.

## 5. Conclusions

In this study, we investigated whether PKM2 can regulate IGF-1R expression and IGF-1R-mediated cancer cell survival, particularly under hypoxic conditions. In addition, we sought to define the molecular mechanisms underlying PKM2-mediated regulation of IGF-1R expression. Our results demonstrate that PKM2 physically interacts with and increases the stability of IGF-1R protein by mediating binding between HSP90 and IGF-1R, whereas PKM2 knockdown decreases the basal level of IGF-1R expression and inhibits AKT activation, the key downstream effector of IGF-1R signaling, consequently increasing apoptotic cancer cell death during hypoxia. Furthermore, we clinically verified the PKM2-regulated expression of IGF-1R from lung cancer patient tissues, which exhibited a significant positive correlation (r = 0.6031, *p* < 0.0001) between PKM2 and IGF-1R expression. Together with the results of our previous report that revealed the direct phosphorylation of PKM2 at serine-202 by AKT, thereby regulating the phospho-S202-PKM2-dependent STAT5 activation, these results suggest that IGF-1R signaling and PKM2 mutually regulate each other to facilitate cell growth and survival, particularly under hypoxic conditions. Moreover, targeting PKM2 represents a promising therapeutic approach for the treatment of solid tumors with dysregulated IGF-1R expression.

## Figures and Tables

**Figure 1 cancers-13-03850-f001:**
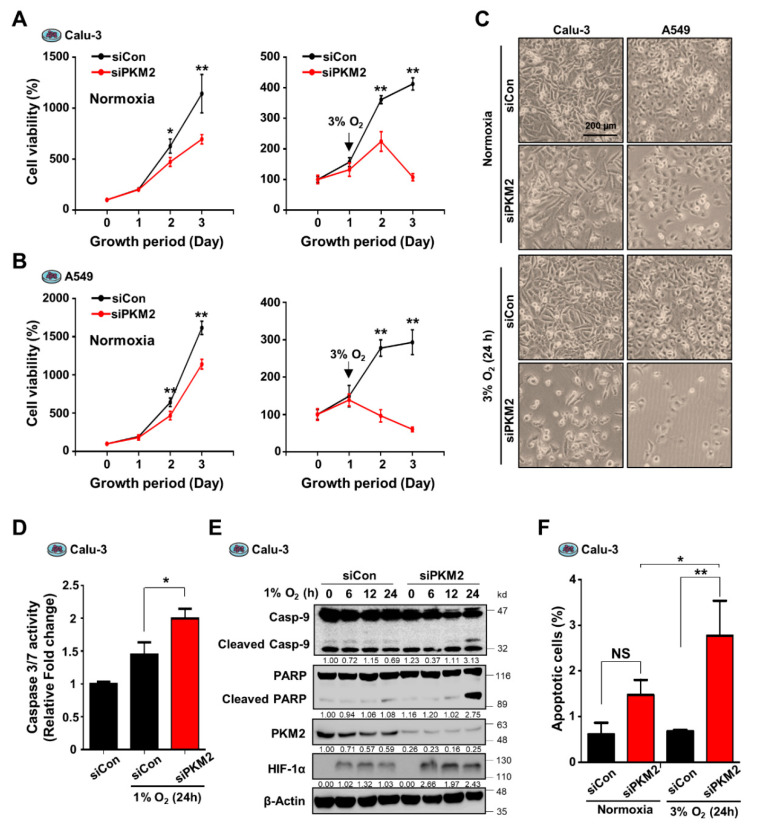
PKM2 is an important regulator of hypoxia-induced apoptosis in lung cancer cells. (**A**,**B**) Cell viability of (**A**) Calu-3 and (**B**) A549 cells transfected with PKM2 siRNA (siPKM2) or Negative Control siRNA (siCon). (Left) Following exposure to normoxic conditions for the indicated time, cells were incubated with CellTiter-Blue® for 2 h. Fluorescence was measured at 530 nm Ex/590 nm Em. (Right) Following incubation in 3% O_2_ for 24 and 48 h, cell numbers were determined by automated cell counter after staining with Trypan blue (*n* = 3). (**C**) Representative images of cells following treatment with PKM2 siRNA captured under an Olympus microscope IX71 with fluorescence and phase contrast. (**D**) Caspase 3/7 activity in Calu-3 cells treated with PKM2 siRNA or Control siRNA. (**E**) Analysis of apoptosis-associated proteins (cleaved Casp-9, cleaved PARP) in Calu-3 cells transfected with PKM2 siRNA or Control siRNA, under hypoxic (1% O_2_) conditions for indicated times. HIF-1α served as an indicator of hypoxic conditions. β-Actin was used as the loading control. (**F**) FACS analysis of Annexin V-stained Calu-3 cells transfected with siCon or siPKM2 (*n* = 3). All values of graphs were presented as mean ± SD. Statistical significance was measured using (A, B, D) one- or (F) two-way ANOVA with the Tukey post hoc test. * *p* < 0.05, ** *p* < 0.01, and NS, statistically not significant.

**Figure 2 cancers-13-03850-f002:**
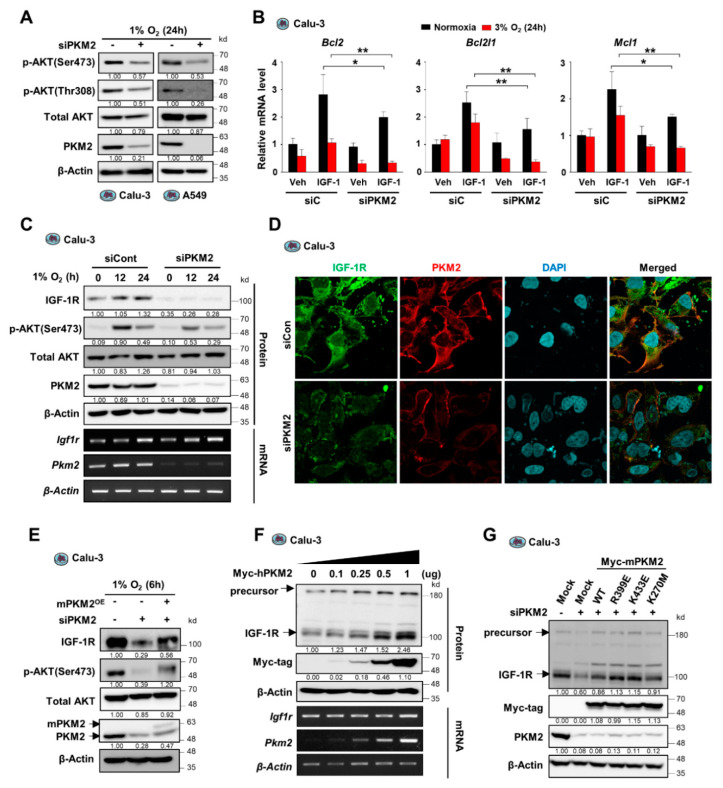
PKM2 deficiency reduces AKT phosphorylation by suppressing IGF-1R expression in hypoxic conditions. (**A**) Abundance of AKT and phospho-AKT proteins in Calu-3 cells transfected with control siRNA or PKM2 siRNA under hypoxic condition (1% O_2_) for 24 h. (**B**) The levels of mRNA expression of anti-apoptotic genes in Calu-3 cells transfected with control siRNA or PKM2 siRNA under normoxic or hypoxic conditions (3% O_2_) for 24 h, normalized to β-actin mRNA. IGF-1 (200 ng/mL) was treated for 24 h. Values of graph were presented as mean ± SD. Statistical significance was measured using one-way ANOVA with the Tukey post hoc test. * *p* < 0.05, ** *p* < 0.01. (**C**) (Upper panel) Protein abundance of IGF-1R, p-AKT(S473), PMK2, and AKT1/2; and (lower panel) mRNA expression for IGF-1R and PKM2 in Calu-3 cells transiently transfected with siPKM2 under hypoxic conditions (1% O_2_) for indicated times. (**D**) Confocal microscopic images of endogenous IGF-1R and PKM2 detected by immunofluorescence in Calu-3 cells transfected with PKM2 siRNA under hypoxic condition (1% O_2_) for 6 h. (**E**) IGF-1R and p-AKT(S473) protein abundance in Calu-3 cells transiently transfected with siPKM2 and/or Myc-tagged mouse PKM2 gene (Myc-mPKM2). (**F**) Protein and mRNA levels of IGF-1R in Calu-3 cells transiently transfected with different doses of Myc-tagged human PKM2 gene (Myc-hPKM2). (**G**) IGF-1R protein abundance in Calu-3 cells transiently transfected with siPKM2 and/or Myc-tagged mutant mPKM2 genes. β-Actin was used as the loading control in (**A**,C,**E**–**G**).

**Figure 3 cancers-13-03850-f003:**
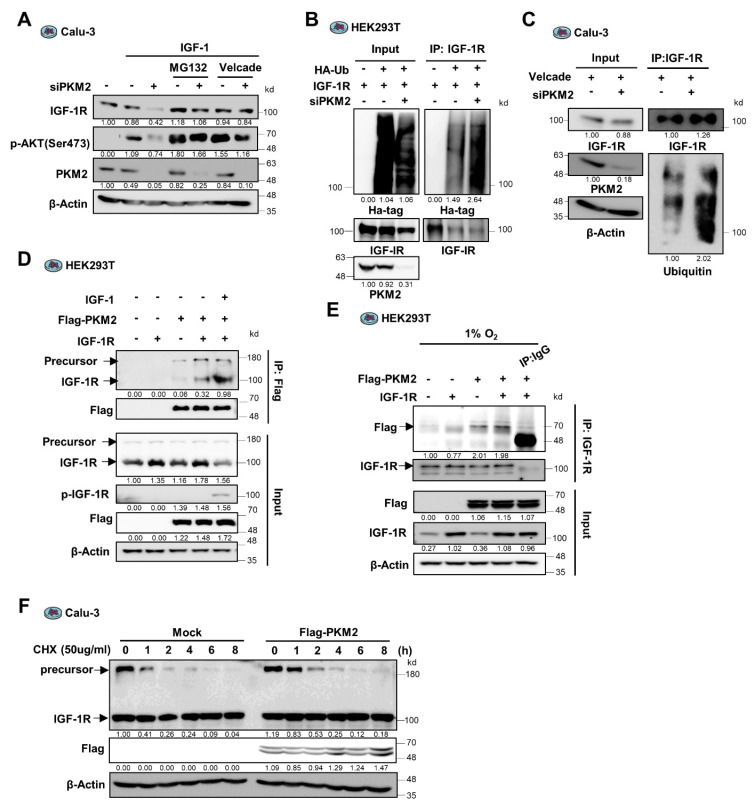
PKM2 physically binds to IGF-1R and regulates its stability via the ubiquitin-proteasome system. (**A**) IGF-1R and p-Ser473-AKT protein abundance in siPKM2-expressing Calu-3 cells pre-treated with the proteasome inhibitors, MG132 (10 μM) or Velcade (10 μM). IGF-1 (200 ng/mL) was treated for 12 h. (**B**,**C**) Ubiquitinated IGF-1R abundance in anti-IGF-1R immunoprecipitates from the extracts of siPKM2-transfected 293 T or Calu-3 cells exogenously expressing IGF-1R and HA-tagged ubiquitin, (**B**) or not (**C**). (**D**) Abundance of IGF-1R precursor and mature proteins in the anti-Flag immunoprecipitates from HEK293T cells exogenously expressing IGF-1R and Flag-PKM2. IGF-1 (200 ng/mL) was treated for 12 h. (**E**) PKM2 protein abundance in anti-IGF-1R immunoprecipitates from HEK293T cells expressing Flag-PKM2 and IGF-1R under hypoxic condition (1% O_2_) for 24 h. Normal mouse IgG was used as the negative control. (**F**) Determination of precursor and mature IGF-1R protein stability in Flag-PKM2 expressing Calu-3 cells following CHX (50 μg/mL) treatment for the indicated time. β-Actin was used as the loading control in (**A**,**C**–**F**).

**Figure 4 cancers-13-03850-f004:**
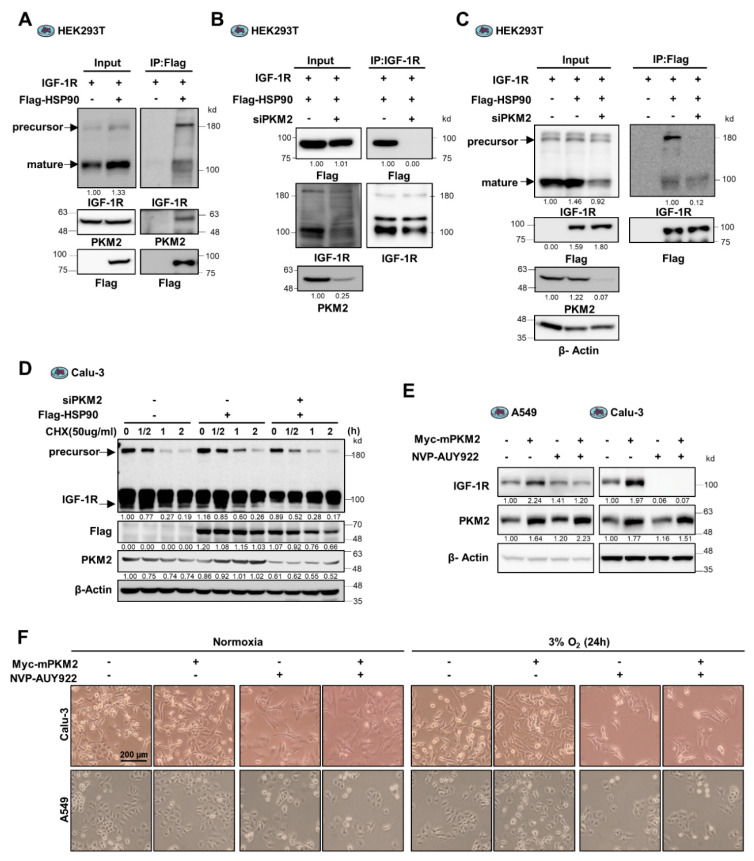
PKM2 regulates the IGF-1R/HSP90 interaction contributing to cancer cell viability. (**A**) Identification of the tertiary complex of IGF-1R, PKM2, and HSP90 proteins in HEK293T cells exogenously transfected with IGF-1R, Myc-PKM2, and Flag-HSP90 genes, as indicated. (**B**) Flag-HSP90 protein abundance in IGF-1R immunoprecipitates from HEK293T cells transfected with control siRNA or PKM2 siRNA. Each cell was also exogenously transfected with IGF-1R and Flag-HSP90 cDNAs, as indicated. (**C**) Abundance of IGF-1R precursor and mature proteins present in Flag-HSP90 immunoprecipitates from HEK293T cells transfected with control siRNA or PKM2 siRNA. Each cell was also exogenously transfected with IGF-1R and Flag-HSP90 cDNAs, as indicated. (**D**) Precursor and mature IGF-1R protein stability assessed following treatment with CHX (50 ug/mL) for the indicated time in Calu-3 cells expressing Flag-HSP90 and/or siPKM2. (**E**) Determination of the effect of NVP-AUY922 (500 nM) treatment on IGF-1R protein abundance in A549 and Calu-3 cells transfected with an expression plasmid for PKM2 or mock vector. (**F**) Determination of the effect of NVP-AUY922 (500 nM) treatment on the viability of Mock or Myc-mPKM2 expressing Calu-3 and A549 cells, cultured under normoxic or hypoxic conditions (3% O_2_) for 24 h. The images were obtained using Olympus microscope IX71 with phase contrast. β-Actin was used as the loading control in (**C**–**E**).

**Figure 5 cancers-13-03850-f005:**
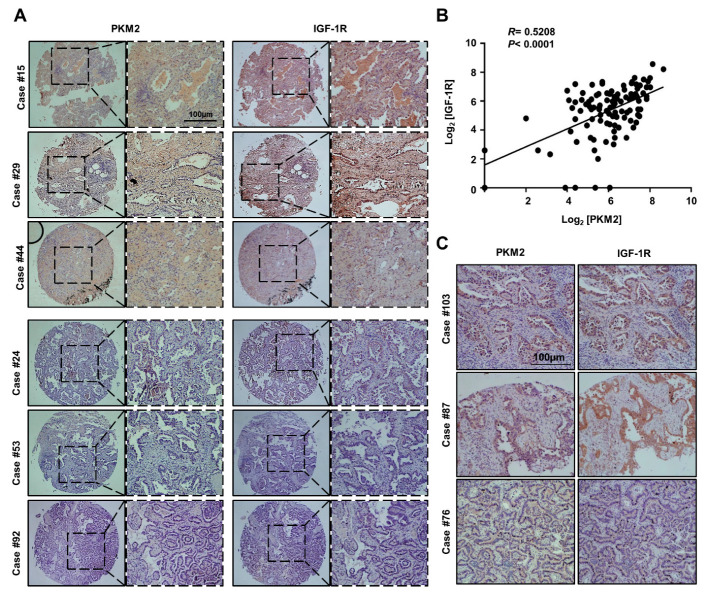
PKM2 expression positively correlates with IGF-1R expression in human lung adenocarcinoma tissues. (**A**) Representative immunohistochemistry (IHC) images of lung cancer tissue microarrays (TMA) using specific antibodies. (**B**) The correlation between IGF-1R and PKM2 protein in lung cancer patient tissues. (**C**) Representative images showing co-localization of PKM2 and IGF-1R expression among the IHC images of the lung cancer TMA.

## Data Availability

Not applicable.

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
