# Peer review of "PKM2 Regulates HSP90-Mediated Stability of the IGF-1R Precursor Protein and Promotes Cancer Cell Survival during Hypoxia"

_cancers, 2021, doi:10.3390/cancers13153850_

Round 1

Reviewer 1 Report

The authors have adequate responses to my comments on the previous version of this manuscript.

Author Response

Responses to the Comments of the Reviewers

Reviewer #1:

The authors have adequate responses to my comments on the previous version of this manuscript.

Response: 

We thank you for your positive feedback.

Reviewer 2 Report

PKM2 plays a crucial role in cancer metabolism and is considered a potential target for anti-cancer therapy. Previous studies were mostly focused on the metabolic activity of PKM2 in cancer. The current manuscript presents the data on the non-metabolic function of PKM2. The authors demonstrated that PKM2 facilitates HSP90 binding to the IGF-1R protein precursor that stabilizes the latter and increases its basal protein amount. The experiments were performed in six cell lines of different origins, with numerous controls. The obtained results are in good agreement indicating that discovered mechanism is universal in many cell types. The manuscript is well written but there are some minor issues:

  1. It is not clear, why the authors used semi-quantitative PCR to estimate mRNA changes instead of quantitative PCR. The method details are not described in the Methods section.
  2. Figure 1E demonstrates the increased level of HIF1a upon PKM2 siRNA treatment. The result should be discussed as it contradicts the cited literature (ref.26) where it is shown that PKM2 activates HIF1a transcription.
  3. The description of the IGF-1 cell treatment is not included in the Methods section and some of the figure captions.
  4. Line 242. It will be better to write “IGF1/Akt signaling-mediated” instead of “Akt signaling-mediated”.
  5. It is not clear, whether the experiments presented on Figs.3B and 3C were performed upon IGF-1R overexpression. Also, the plasmids and transfection conditions used for overexpression experiments were not described in the Methods section.
  6. The numbers of tissue microarray samples are different in the Abstract, Methods, and Results sections. This requires editing or clarification.
  7. Figure 5B shows the correlation between IGF-1R and PKM2 protein expression. For most of the samples, the presented values are of a small magnitude. It is worth exploring the scatterplot and the correlation in the log-log scale (and, additionally, explicitly report the number of zeros).

Author Response

Responses to the Comments of the Reviewers

Reviewer #2:

PKM2 plays a crucial role in cancer metabolism and is considered a potential target for anti-cancer therapy. Previous studies were mostly focused on the metabolic activity of PKM2 in cancer. The current manuscript presents the data on the non-metabolic function of PKM2. The authors demonstrated that PKM2 facilitates HSP90 binding to the IGF-1R protein precursor that stabilizes the latter and increases its basal protein amount. The experiments were performed in six cell lines of different origins, with numerous controls. The obtained results are in good agreement indicating that discovered mechanism is universal in many cell types. The manuscript is well written but there are some minor issues.

Comment 1:

It is not clear, why the authors used semi-quantitative PCR to estimate mRNA changes instead of quantitative PCR. The method details are not described in the Methods section.

Response: 

We apologize for the insufficient explanation. A more detailed description of the PCR conditions has been provided in the Materials and Methods section (lines 140-151). We used semi-quantitative PCR to estimate the mRNA levels when it was necessary to compare against protein expression levels. When more accurate quantification of mRNA expression was required as shown in Figure 2B, we used quantitative PCR.

Comment 2:

Figure 1E demonstrates the increased level of HIF1a upon PKM2 siRNA treatment. The result should be discussed as it contradicts the cited literature (ref.26) where it is shown that PKM2 activates HIF1a transcription.

Response: 

We apologize for the confusion in citing the references. We accidently omitted the reference for the non-metabolic function of PKM2 to promote HIF-1É‘ activity (ref.18) in the Results section. According to this study, PKM2 promotes HIF1 activity by enhancing HIF1 binding and p300 recruitment to hypoxia response elements, and not HIF1 mRNA or protein expression. The cited literature (ref.26) is for the non-metabolic function of PKM2 function to phosphorylate histone H3 at Thr-11. We have added the reference for the non-metabolic function of PKM2 to promote HIF-1É‘ activity (ref.18) (line 259).

Comment 3:

The description of the IGF-1 cell treatment is not included in the Methods section and some of the figure captions.

Response:

We apologize for the insufficient explanation. We have added information about IGF-1 used in the study in the Materials and Methods section (lines 99-100). Treatment time and concentration of IGF-1 have been described in the captions of Figures 2 and 3.

Comment 4:

 Line 242. It will be better to write “IGF1/Akt signaling-mediated” instead of “Akt signaling-mediated”.

Response: 

We thank you for the suggestion. We have made the necessary revision accordingly (line 267).

Comment 5:

It is not clear, whether the experiments presented on Figs.3B and 3C were performed upon IGF-1R overexpression. Also, the plasmids and transfection conditions used for overexpression experiments were not described in the Methods section.

Response: 

We apologize for the unclear explanation. For clarity, we have revised the explanation of Figure 3B and 3C, to improve clarity, in the result section (line 309) and figure caption. The information about plasmid and transfection conditions has been added in the Materials and Method section (lines 122-138).

Comment 6:

The numbers of tissue microarray samples are different in the Abstract, Methods, and Results sections. This requires editing or clarification.

Response: 

We apologize for the inconsistency. The description of the number of samples in the Materials and Methods (line 207) and the Results section (line 381) were corrected.

Comment 7:

Figure 5B shows the correlation between IGF-1R and PKM2 protein expression. For most of the samples, the presented values are of a small magnitude. It is worth exploring the scatterplot and the correlation in the log-log scale (and, additionally, explicitly report the number of zeros).

Response:

We thank you for this helpful comment. Following the suggestion, we have changed the value in the scatterplot in Figure 5B to the log-log scale. To change the expression scores at the linear scale to the log scale, we added a score “1” to the expression scores of all samples including the 7 samples with zero score, and then converted to the log scale. We have made the corresponding revisions in the Abstract (line 37) and the Results sections (line 385).

We thank the reviewers for helping us strengthen the manuscript with insightful comments and suggestions.

This manuscript is a resubmission of an earlier submission. The following is a list of the peer review reports and author responses from that submission.

Round 1

Reviewer 1 Report

Comments and Suggestions for Authors:

The manuscript identifies that the regulation of IGF-1R expression by PKM2, suggesting a positive feedback regulation of IGF-1R signaling by PKM2 in hypoxic conditions. The authors investigated that the downregulation of PKM2 leads to the attenuation of IGF-1R signaling-mediated phosphorylation of AKT via the suppression of IGF-1R protein expression and the promotion of hypoxia-induced apoptosis. Although the paper is interesting, it also includes several problems as described below.

Comment 1:

Pyruvate kinase (PK) is the enzyme involved in the final step of glycolysis, catalyzing the dephosphorylation of phosphoenolpyruvate (PEP) to pyruvate and generating an

adenosine triphosphate (ATP) molecule [PMID: 23773105]. Please the authors need to provide justification for why PKM2 protein kinase, but not another gene. In addition, are there any previous studies on the regulation of PKM2 influence cancer cells through different signaling pathways by micro-RNAs in hypoxia? If so, please the authors need to cite the references.

Comment 2:

Besides transcriptional regulation, Hsp90 is regulated by posttranslational modifications (PTMs), interaction with Hsp90 cochaperones, and surprisingly, even by binding to clients. Numerous PTMs including small ubiquitin-like modifier addition (SUMOylation), acetylation, phosphorylation, and S-nitrosylation have been described and reviewed before. Can the authors propose/provide other protein targets that might be more directly regulated by Hsp90, especially at posttranslational modifications?

Comment 3:

There is a lack of information particularly in the materials and methods section, specifically concerning the recruitment of patients. Is there any informed consent about the patients included in the study?

Comment 4:

It is suggested to use more updated references in this manuscript.

Reviewer 2 Report

The manuscript by Koo et al is a generally well-presented paper. Yet, I have the following concerns.

  1. Please mention more about the method for establishing the hypoxia environment in Materials and Methods.
  2. There is a mismatch of Figure with Figure legends (Fig 4,5)
  3. Fig 1, why some parts used 1% O2 and some used 3% O2?
  4. Fig 2D, unclear image, please replace.
  5. Fig 2, most of the figure panels presented one single cell line, please add the data of at least one more cell line. For example, all figure panels will have Calu-3 and HEK293T.
  6. Fig 3, most of the figure panels presented one single cell line, please add the data of at least one more cell line.
  7. Fig 4, similarly, most of the figure panels presented one single cell line, please add the data of at least one more cell line.
  8. Fig 5, resolution too low, please update the figures.